# No Change, No Life? What We Know about Phase Variation in *Staphylococcus aureus*

**DOI:** 10.3390/microorganisms9020244

**Published:** 2021-01-25

**Authors:** Vishal Gor, Ryosuke L. Ohniwa, Kazuya Morikawa

**Affiliations:** 1Graduate School of Comprehensive Human Sciences, University of Tsukuba, Tsukuba, Ibaraki 305-8575, Japan; 2Faculty of Medicine, University of Tsukuba, Tsukuba, Ibaraki 305-8575, Japan; ohniwa@md.tsukuba.ac.jp

**Keywords:** *Staphylococcus aureus*, phase variation

## Abstract

Phase variation (PV) is a well-known phenomenon of high-frequency reversible gene-expression switching. PV arises from genetic and epigenetic mechanisms and confers a range of benefits to bacteria, constituting both an innate immune strategy to infection from bacteriophages as well as an adaptation strategy within an infected host. PV has been well-characterized in numerous bacterial species; however, there is limited direct evidence of PV in the human opportunistic pathogen *Staphylococcus aureus*. This review provides an overview of the mechanisms that generate PV and focuses on earlier and recent findings of PV in *S. aureus*, with a brief look at the future of the field.

## 1. Introduction

The Gram-positive human commensal *Staphylococcus aureus* is an opportunistic pathogen that imposes a major health and economic burden on a global scale [1]. *S. aureus* can colonize multiple sites of the human body, but the primary niche of commensal colonization is the anterior nares, with various other skin surfaces making up secondary niches. There are three main carrier-patterns of *S. aureus* amongst healthy individuals: persistent carriers (~20%), intermittent carriers (~30%), and non-carriers (~50%) [2]. Nasal carriage of *S. aureus* has been linked to a higher chance of contracting infection [2]. *S. aureus* is responsible for an astounding diversity of infections. It is the leading cause of infective endocarditis, osteoarticular infections, and surgical site infections and *S. aureus* bacteraemia is also prevalent [3,4]. *S. aureus* can also cause pneumonia and other respiratory infections, particularly in people living with cystic fibrosis [3]. Furthermore, *S. aureus* is supremely adept at colonizing alien surfaces within the body and is often responsible for infections associated with catheters, cannula, artificial heart valves, and prosthetic joints [3]. This diverse range of infections is enabled by a vast arsenal of virulence factors that are ready to be deployed in a variety of host environments [5,6]. Of particular concern is *S. aureus’* rapid development of antibiotic resistance. Methicillin Resistant *S. aureus* (MRSA) has broad-spectrum resistance against the β-lactam group of antibiotics and is a global danger with clones existing in both nosocomial and community settings [7]. MRSA is also a problem in the livestock sector, where it can co-infect both animals and humans [8]. The infamous development of antibiotic resistance, coupled with its worrying genetic plasticity, has earned *S. aureus* a place in the ESKAPE group of pathogens: a collection of bacteria that represent paradigms of acquisition, development, and transfer of antibiotic resistance [9]. Thus, to better combat this dangerous pathogen it is vitally important to study adaptation mechanisms of *S. aureus*.

Another particular trait of *S. aureus* that makes it notoriously difficult to combat in the clinical setting is phenotypic heterogeneity. An example of this is the phenomenon of persister cells, where sub-populations of *S. aureus* gain a resistance phenotype against antibiotic treatment resulting from arrested growth [10]. Persister cells may be generated in numerous ways, one of which is the formation of Small Colony Variants (SCVs) that are characterized by auxotrophy for various compounds involved in the electron transport chain and slow growth, allowing them to escape the effects of many antibiotics [11,12]. Importantly, these populations do not acquire conventional resistance mechanisms against the antibiotics. This heterogenous phenomenon has severe clinical implications and is thought to be a significant cause of antibiotic treatment failure and chronic recurrent infections [13].

Heterogeneity is not limited to antibiotic resistance. As we discuss in this review, diverse traits, including pathogenicity factors, have recently been recognized as having sub-population patterns of expression. The scientific point-of-view has been increasingly focused on such heterogenous phenomena, yet progress is still in the relatively early stages and much work remains to be done. In this review, we summarize the information regarding bacterial Phase Variation (PV), a mechanism of high-frequency reversible gene expression switching (Section 2) and collate the known examples of *S. aureus* PV into one source (Section 3) to aid in future studies on heterogeneity in *S. aureus* (Section 4).

## 2. Bacterial Phase Variation

### 2.1. Background of Phase Variation

All living organisms are faced with the constant challenge of maintaining fitness in order to survive and reproduce, and this is no less true for bacterial species. Bacteria are under constant onslaught from fluctuations in their local environment, infection from bacteriophages, and (in the case of pathogenic bacteria) attack from their infected host. Although bacteria possess robust mechanisms of classical gene regulation that allow them to respond to extracellular changes (e.g., Bacterial Two-Component Systems), these alone may be unable to cope with the constant barrage of fluctuating pressures they face. These selective pressures are often focused on bacterial external proteins which form the first line of contact with the outside environment and this has led to development of what have been termed “contingency loci” [14,15]. Contingency loci are hypermutable genes that generate genetic and phenotypic variation allowing bacterial populations to survive unpredictable pressures. This hypermutability is conferred by the phenomena of Phase Variation (PV) and antigenic variation.

PV is a reversible gene expression switch that can alter expression between an ON and an OFF state and occurs through several genetic and epigenetic mechanisms [16]. It is characterized by high frequencies, usually exceeding 1 × 10^−5^ variants per total number of cells [17,18] which is orders of magnitude above the typical frequencies of spontaneous mutations (10^−6^ to 10^−8^ per cell per generation) [18]. Depending on the method of calculation, the frequency of PV may describe not only rate of the PV mechanisms but also the growth of the phase variants themselves. Antigenic variation is related to PV and occurs through similar mechanisms. However, rather than alternating between an ON and OFF state, antigenic variation mechanisms generate variations in the sequence of surface proteins resulting in the expression of different forms and structures of the antigenic proteins on the cell surface [17,18,19]. As such, due to the similar nature of the mechanisms involved, antigenic variation will not be separately addressed in this review.

As mentioned above, genes subject to PV often encode for cell-surface associated features such as adhesins, liposaccharide synthesis enzymes, and pili [20,21,22] but can also encode for virulence factors and secreted proteins such as iron acquisition machinery [23,24]. The collection of phase variable loci in a bacterial species is referred to as the “phasome” [16] and generally includes genes which are involved in bottlenecks experienced by the bacterial population. This is most clearly seen in pathogenic bacteria which undergo constant challenge from host immunity during the infection process. For example, PV mediated shutdown of liposaccharide synthesis genes in the invasive pathogen *Haemophilus influenzae* confers protection against neutrophil-mediated immune clearance but is detrimental in other environments [22,25,26]. In another example, PV in *Salmonella typhimurium* flagellae can modulate their antigenic properties and allow for evasion from host immunity [27]. 

It is likely that the original role of PV was as a mechanism of innate immunity against bacteria’s greatest enemy: bacteriophages [28]. Although bacteriophages exist in exaggerated abundance relative to their bacterial hosts, their host range is often limited to just a few specific strains of a given bacterial species [29]. Thus, there is a constant cyclical arms race between bacteria and bacteriophages in order to stay one step ahead of each other [30], and PV plays an important role in both sides of this war. An example can once again be found in liposaccharide synthesis genes of *H. influenzae* in which PV can result in a switch from a sensitive to resistant phenotype against the HP1c1 phage [31]. On the other hand, PV in the *Escherichia coli* phage Mu causes a switch in expression between two sets of tail fibers resulting in modulation of the host specificity [32,33] with similar phenomena identified in other phages [34].

Considering the above information, it can be inferred that genetic loci susceptible to PV would be found in abundance amongst bacterial species that experience population bottlenecks. Typically, such bottlenecks often occur during the infectious process which imposes limits onto the bacterial population size. These bottlenecks reduce genetic diversity at a time when variation is most beneficial, and PV offers a solution to this hurdle and indeed, several pathogenic bacteria have been documented to undergo PV [18]. 

While PV is, by definition, a stochastic process, it occurs through several discrete mechanisms. Broadly speaking, mechanisms of generating PV can be discriminated into genetic and epigenetic mechanisms [16] both of which will be addressed in Section 2.2 and Section 2.3 respectively.

### 2.2. Genetic Mechanisms of PV

There are three genetic mechanisms of PV which shall be discussed in the following chapters: Variation in length of DNA Short Sequence Repeats (SSRs) [35,36,37], DNA inversion [38], and DNA recombination [39,40]. 

#### 2.2.1. Variation in Length of DNA Short Sequence Repeats (SSRs)

SSRs are homo- or hetero-nucleotide repeats in DNA that are highly prone to insertion/deletion (indel) errors due to Slipped-Strand Mispairings (SSMs) during DNA replication [35,36,37]. SSRs can be as complex as repeating units of tetranucleotides or as simple as a straight homonucleotide run. Indels in SSRs can result in frameshifts that largely have an ON↔OFF effect on protein function or gene expression (by resulting in abrupt termination of translation or inhibition of RNA polymerase binding, respectively Figure 1A) but can have an alternative gradation effect on gene expression as well. For example, alterations in the length of a dinucleotide TA10 tract in the promoter regions of the divergently transcribed *hifA* and *hifB* genes controlling fimbriae expression in *H. influenzae* can either significantly affect *hif* expression (TA10→TA9) or only moderately affect it (TA10→TA11) [41]. The evolution of the mutability of SSR tracts is largely driven by a combination of environmental and molecular drivers. The environmental drivers include factors such as the aforementioned population bottlenecks arising during infection processes. These bottleneck conditions exert a primary selective pressure for phenotypes that can survive them, e.g., a population that can shut down the expression of a surface protein that is targeted by host immunity. The necessity to survive this recurrent primary selection serves as a secondary layer of selection for plasticity of the gene itself.

The molecular factors are intrinsic to SSR tracts and include the DNA replication and the Mismatch Repair (MMR) [42]. The discriminating factors of SSRs can be broadly delineated into two groups: the composition of the repeating nucleotide unit (i.e., a homonucleotide or a heteronucleotide repeat) and the tract length. These in turn are differentially affected by the DNA replication and MMR machinery. Amongst these proteins are the DNA polymerase enzymes which include the polymerase responsible for the construction of new DNA strands (DNA polymerase III) as well as the polymerase responsible for DNA repair (DNA polymerase I). Studies have shown that these polymerases have an inherent frequency of generating addition/deletion errors when constructing new DNA strands [43,44]. Following DNA replication, any errors are corrected by the MMR machinery which is a suite of Mut proteins that target and fix errors in a strand specific manner. Inactivation of components from either of these suites of proteins results in a hypermutable phenotype and can lead to SSR alteration e.g., [45]. Additionally, the hypermutable phenotype that results from loss of the MMR machinery is directly responsible for genetic variability of bacteria and mutator phenotypes play an important role in bacterial adaptation [46]. For example, both *S. aureus* and *Pseudomonas aeruginosa* isolated from the lungs of people suffering from cystic fibrosis are commonly associated with antibiotic resistance caused by hypermutability [47,48,49]. Interestingly, while both the MMR machinery and DNA polymerases are involved in SSR evolution, they do not appear to be fully redundant. Several studies have shown that MMR is more responsible for variability of homonucleotide SSRs, especially for those which exceed eight nucleotides in length, whereas DNA polymerase I is exclusively responsible for mutations in heteronucleotide SSRs [50,51,52]. This could have evolutionary implications for the mechanisms of generating SSRs. For example, *H.influenza* is enriched with tetra-nucleotide SSRs [51] whose expansion/contraction is affected by DNA polymerase I. Furthermore, evidence suggests that the frequency of DNA polymerase I mediated errors differs between the leading and the lagging strands of newly synthesized DNA, implying that the direction of genes in the chromosome can also dictate the type of SSR that would evolve in them [53]. Lastly, an interesting study carried out by Lin et al. investigated the distribution of SSRs within the genomes of several bacterial species. They found that in many pathogenic species, SSRs were enriched towards the N-termini of protein coding sequences increasing the probability of frameshifts resulting in non-functional proteins [54,55]. This further suggests that bacteria have evolved SSRs in a manner to provide maximal PV. 

#### 2.2.2. DNA Inversion

DNA inversion was the first documented example of PV, though the mechanism was not known at the time the phenomenon was documented [38] (Figure 1B). It involves recognition of inverted repeat (IR) sequences by invertase enzymes and subsequent enzyme-mediated inversion of the DNA. An elaborate study was carried out by Jiang and colleagues who developed an algorithm to search published bacterial genome datasets for IR sequences that might be phase variable [56]. Not only did they identify that IR sequences were enriched in host-associates species (implying a benefit of PV during commensalism or infection) but they also discovered three antibiotic resistance genes regulated by invertible promoters: a macrolide resistance gene, a multidrug resistance cassette conferring resistance to macrolides and cephalosporins, and a cationic antibacterial peptide resistance operon [56]. The presence of antibiotics influenced the switch from an OFF to an ON state for these genes. Some of the invertible promoters seem to be located on genetic elements homologous to those conveyable by horizontal gene transfer mechanisms, raising the worrying possibility that these resistance gene switches can be transferred to other species [56].

#### 2.2.3. DNA Recombination

Homologous recombination provides a pathway for DNA re-arrangement and subsequent PV. Events arising from recombination mechanisms are often due to DNA deletions, and thus tend to be in a one way ON→OFF direction. However, gene duplication or transfer events can often occur to balance out the accumulation of inactive variants in the population. A well characterized example of recombination mediated variation occurs in the *Neisseria gonorrhoea* pilus organelle, which is essential for full infectivity and natural transformation. *N. gonorrhoea* contains a *pilE* gene that encodes for a pilin protein that is the major component of the pilus, but also contains several silent *pilS* alleles several Kb away [39]. RecA-dependent recombination events can unidirectionally transfer large sections of the *pilS* allele into *pilE*, thus creating an OFF variant [40,57] (Figure 1C). The *N. gonorrhoea* pilus also undergoes PV by SSM-mediated variation in the length of a poly(G) tract in the *pilC* gene (which encodes for the adhesive tip of the pilus [58]) resulting in ON↔OFF switching [59,60].

### 2.3. Epigenetic Mechanisms of Phase Variation

An epigenetic trait has been defined as a heritable phenotype resulting from modified gene expression that is not due to any alterations in the DNA sequence of the chromosome [62,63]. In prokaryotes, DNA methylation occurs mainly at the nucleotide adenine although studies have shown that cytosine methylation can also occur [64,65,66]. DNA methylation usually occurs at specific target sites and is carried out either by methyltransferases that are part of dedicated Restriction–Modification (RM) systems or by orphan methyltransferases. A well-studied methylase responsible for bacterial epigenetic regulation of PV is the DNA Adenine Methyltransferase (DAM) which is an orphan methyltransferase of the gammaproteobacterial family that is specific for GATC sites [64]. Methylation of DNA represses transcription, and thus PV can result if there are GATC sites within a gene promoter which also binds transcription factors, causing mutually exclusive binding competition between the transcription factor(s) and DAM. If there are numerous GATC sites within a promoter region then the mutually exclusive competition can result in differential methylation patterns of the promoter region resulting in switching between an ON and OFF state. A paradigm of this sort of PV is established by a series of intriguing reports studying the *pap* operon of *E. coli* and the *opvAB* operon of *Salmonella enterica* [21,67,68,69].

### 2.4. Combined Mechanisms of Phase Variation

There is growing evidence that shows that many bacterial species undertake a combined approach for PV to maximize the ability to generate rapid and diverse variation. This strategy involves generating PV through genetic mechanisms in genes of RM systems that can modify the transcriptome of the cell via epigenetic control. Such systems are referred to as “phasevarions” as they control phase-variable regulons [70] and are immensely powerful weapons in the arsenal of pathogens. 

The earliest phasevarions identified are controlled by Type III RM systems. PV occurs in SSRs in the *mod* gene resulting in ON↔OFF variation and altered methylation states [71,72]. Strikingly, analyses of known Type III system sequences indicate that at least 20% of these systems contain SSRs and could potentially be phasevarions [73]. Furthermore, *mod* genes are highly conserved, with variation occurring mainly in the DNA recognition domain. This allows *mod* genes to exist within the species as multiple alleles, each of which controls distinct phasevarions [16]. 

There is some evidence of a Type II RM regulated phasevarion detected in *Campylobacter jejuni*, and gene expression patterns were detectably different upon RNAseq analysis, though no direct link to any altered phenotype was reported [74].

PV in Type I systems largely occurs through DNA inversion in the *hsdS* gene, creating multiple allelic variants of the specificity protein of the Type I system resulting in different gene targets upon PV [75]. An example of a Type I RM phasevarions can be seen in variable capsular expression controlling virulence in *Streptococcus pneumonia* [16,76].

In theory, phasevarions must be also seen by PV in other regulators of gene expression, such as transcription factors. Some examples of these are described in Section 3.

## 3. Known Examples of Phase Variation in *S. aureus*

*S. aureus* is an opportunistic pathogen that has claimed several distinct niches in the human body, thus being subject to a variety of different conditions and stresses against which it has accumulated diverse colonisation and pathogenicity factors. As such, it is primed to exploit the phenomenon of PV; however, there have been surprisingly few documented reports of PV examples, possibly due to a link between heterogenous phenotypes and PV not being made. This section (Section 3) will outline reports that have identified PV, or PV-like mechanisms, in *S. aureus*. Reports detailing heterogeneity arising from non-PV mechanisms will not be discussed as they have been detailed elsewhere [77].

Perhaps the best studied example of PV in *S. aureus* relates to its ability to form biofilms. A report from the early 1990s identified phase variation in the production of an extracellular polysaccharide coat (or “slime”) whose production could be reversibly switched across generations of the same lineage with variants easily distinguishable by differential colony morphology on Congo Red Agar [78]. From there, the topic took on multiple approaches from various groups. Tormo and colleagues identified that expression of the Bap protein (a major surface component involved in biofilm formation that promotes primary attachment as well as intercellular adhesion) was phase variable [79]. However, although they confirmed that Bap-negative variants did not express the *bap* gene, they could not detect any sequence alterations, suggesting that the exact mechanism of PV was either indirect or occurred through epigenetic means. Investigating from another direction, Valle and colleagues discovered that *IS256* transposition can also result in biofilm PV by disrupting the *sarA* regulator and *icaC* [80]. The *icaADBC* operon encodes for genes involved in the synthesis of poly N-acetylglucosamine exopolysaccharide and its deacylated variant polysaccharide intercellular adhesion (PNAG/PIA) [81](other major extracellular components involved in biofilm formation that also have roles in immune evasion [82]). Interestingly, they further discovered that there is a connection between this variation and the global stress sigma factor σ^B^, as a σ^B^ deletion mutant has significantly higher *IS256* copies and transposition frequencies [80]. However, there are yet more layers to this example. In 2003, Jefferson and colleagues discovered a 5-nucleotide SSR (TATTT) in the promoter region of the *ica* operon, whose expansion/contraction affected the binding of *ica* regulatory elements and shut down PNAG/PIA production [83]. In a subsequent study, Brooks and Jefferson discovered that there are further SSRs present in the operon in the form of tetranucleotide repeats within the *icaC* ORF (Open Reading Frame), and SSM events in those also reversibly control PNAG/PIA production [81]. Finally, an elaborate mechanism for the *icaC* SSM expansion has been proposed. The *icaC* tetranucleotide SSR can stably form a so called “mini dumbbell structure” by folding back on itself and making a small loop [84]. It has been proposed that if such a structure were to form during DNA replication it would increase the frequency of SSM events, resulting in expansion of the SSR [84].

Surface proteins are theoretically particularly prone to PV, and one of the first conclusively identified PV events occurs in the extracellular MapW protein, which may have functions in immune evasion based on its high degree of similarity with Major Histocompatibility Complex Class II molecules [85]. The *mapW* gene has a poly(A) tract that results in premature termination of translation. A change in the poly(A) tract can shift the reading frame and result in full-length protein being transcribed [85]. This example of PV varies the length of the protein product rather than switching between an ON↔OFF state, though it is yet to be confirmed if both the truncated and full-length protein have distinct functions.

A PV-like system was also found in the regulation of natural transformation just short of a decade ago [86]. The finding of staphylococcal natural transformation has implications regarding its rapid acquisition of antibiotic resistance genes, but intriguingly it was demonstrated that only a subset of the population can enter a competence state. The genes necessary for entering the competence state are controlled by the alternative sigma factor σ^H^ [87] and the transcription factor ComK that synergistically works with σ^H^ [88]. It was found that two independent mechanisms control σ^H^ expression in *S. aureus* [86]. The translation of the σ^H^ mRNA is likely repressed through the action of an inverted repeat loop in the 5’ UTR, and the still elusive de-repression mechanism allows σ^H^ expression in subpopulations. In addition, as a second genetic mechanism, at low frequencies (~≤10^−5^) *sigH* undergoes a gene duplication event with downstream genes, effectively replacing the native 5’ UTR, and thus lifting repressive control [86]. This event is reversible and reverts to the native chromosomal structure at a frequency of 10^-2^. Although the frequency of duplication is lower than that commonly associated with PV, the reversible nature of the mechanism coupled with the contingency-like nature of the *sigH* locus [89] allows for a justification of this phenomenon being discussed under the umbrella of PV.

A very recent study in our lab uncovered another example of PV in *S. aureus*, one with potentially far-reaching implications. Upon investigating the phenomenon of hemolytic heterogeneity commonly observed in *S. aureus*, we identified PV-controlled reversible shutdown of the central virulence regulatory system, the Accessory Gene Regulator (Agr) system [90]. PV occurred via two distinct mechanisms: the first was a duplication and inversion event within the ORF of *agrC* (which encodes for the sensor component of the Agr TCS) and the second involved alteration in the length of homonucleotide SSRs within the *agrA* ORF (which encodes for the TCS response regulator). The second mechanism was also identified in a single clinical isolate, and although we were unable to determine the clinical significance of this findings (owing to a minor frequency of clinical revertant strains), the results demonstrated that *S. aureus* has a phasevarion under control of the Agr system. Furthermore, a study that investigated *S. aureus* dermal colonization in children identified that chronic colonizers tended to have a mutationally inactivated Agr system. Importantly, we found that two of their samples (out of four) had frameshifts resulting from alterations in homonucleotide SSRs within the *agrC* and agrA ORFs [91]. The implications of this suggest that this phasevarion could come into play as a cryptic insurance strategy against host-mediated immune attack and may possibly even allow *S. aureus* to manipulate host phagocytic cells and use them as a Trojan horse to disseminate itself within the host (Figure 2). This is of particular consideration as under certain conditions bacterial infections can be established by a single surviving cell [92].

## 4. Future Perspectives

There is mounting evidence of PV being important in the evolution and adaptation of bacteria, with roles ranging from their arms race with phages to their pathogenic proficiency. *S. aureus* is an important global pathogen that can survive in a variety of different niches within the human body using its arsenal of virulence factors. Considering this, *S. aureus* should be primed to exploit PV in its lifestyle yet there remain few documented cases of clear PV phenomena. However, it is very possible that cases reminiscent of PV have been overlooked in the past and may be worth further investigating. For example, a study carried out by Aarestrup and colleagues as far back as 1999 [95] documented heterogenous expression of the alpha and beta hemolysins of *S. aureus* amongst strains that carried the hemolytic genes. This could be due to PV shutdown of these hemolysins in the non-expressing strains. We recently identified non-hemolytic clinical isolates that could revert hemolytic activity without any change in their Agr phenotype [96], which could support the idea that the heterogenous hemolytic phenotypes observed by Aarestrup arise from PV. 

With modern developments in genetic and experimental technology, the way we can go about investigating phenomena such as PV has drastically changed. This is no more clearly demonstrated by Jiang et al. who mined genome databases for inverted repeat regions as a primary screen for potentially phase variable genes [56]. In an attempt to gain similar preliminary insight, we screened the genome of the highly virulent community-acquired Methicillin-Resistant strain MW2 for genes that contained homonucleotide SSRs of adenine and thymine that are 6 nucleotides or longer within the ORF and the putative promoter region of the gene. We focused on homonucleotide tracts as three cases of PV in *S. aureus* have been demonstrated to occur through homonucleotide tract-length alteration (*mapW* and 2 discrete SSRs in *agrA*, [85,90]). Our initial results were astonishing. More than 700 genes contained at least one Poly(A) or Poly(T) SSR, with a substantial number containing 3 to 4 SSRs, and one gene containing a stunning 26 SSRs (Appendix A). These initial data corroborate an extensive study carried out by Orsi et al. who identified that Poly(A)/Poly(T) tracts are overrepresented in numerous bacterial genomes [54]. Interestingly, we noticed a greater abundance of Poly(A) tracts compared to Poly(T) in coding sequences. This is corroborated by Orsi et al., who found a similar result when looking at tracts longer than 6 nucleotides in length, though the difference reduced with shorter tracts. Surprisingly however, although Orsi et al. identified that these SSRs are predominantly located near the 5’ end of the CDS (coding sequence), we only observed this distribution pattern for Poly(T) SSRs, not Poly(A) (Figure 3). Taken together, PV in *S. aureus* may be severely under-reported, which is understandable as most PV may not be noticeable through conventional bulk analysis and elaborate experimental systems may be needed to observe the PV-mediated switch in gene expression. Intriguingly, approximately 13% of genes (from those with known function) that contained Poly(A) or Poly(T) SSRs were essential genes. If these genes are indeed subject to PV, it raises a question as to what circumstances or trade-offs could possibly merit the shutdown of essential genes as advantageous. Transcriptional slippage at Poly(A)/Poly(T) tracts can lead to a population of mRNAs with varying tract length, some of which may contain the correct number of nucleotides for the entire CDS to be in-frame [97]. This could enable low-level expression of essential genes that have been shut down by PV. Alternatively, the phenomenon of biological hysteresis [98], wherein functional proteins are inherited by daughter cells during splitting from the mother cell, could support daughter cells for a short period without further de novo synthesis, potentially giving chances of further changes to the SSR.

The aim of this review is to stimulate interest in identifying PV in *S. aureus* and to increase attention to an area of study that warrants more investigation with modern technological approaches. Here, we have described a comprehensive understanding of the mechanisms of PV and the scope of the discoveries yet to be made in *S. aureus.* Further investigations focusing on PV in *S. aureus* are sure to lead to exciting new information, and the more we learn of the ingenious adaptation mechanisms this important pathogen employs during its infectious process, the better we will be equipped in dealing with it.

## Figures and Tables

**Figure 1 microorganisms-09-00244-f001:**
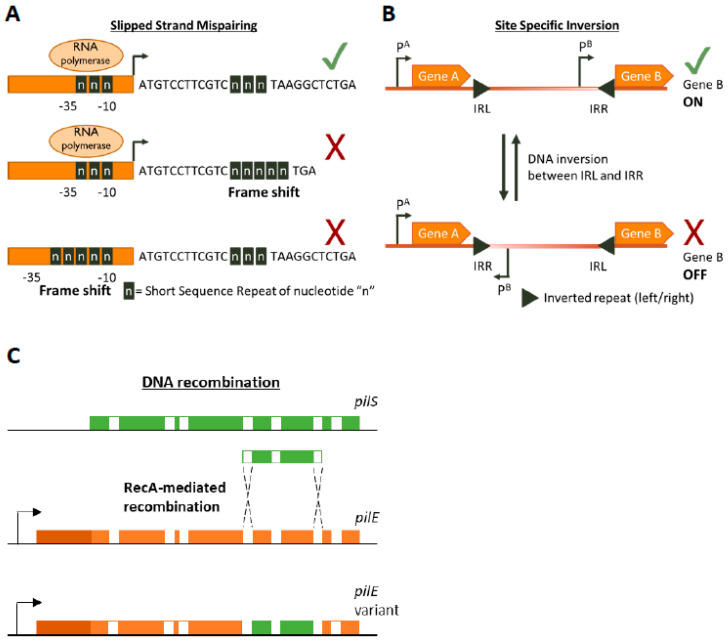
Genetic mechanisms of Phase Variation. A cartoon depicting the three main genetic mechanisms of Phase variation (PV). (**A**) Slipped-Strand Mispairing events within Short Sequence Repeats (SSR) result in expression (green tick mark) of truncated dysfunctional proteins (if SSR is in the CDS) or inhibition (red cross) of transcription by preventing RNA polymerase/transcription factor binding or by other mechanisms. For example, an interesting method of PV-mediated transcriptional control is shown by Danne et al. who demonstrate SSR alterations upstream of the *pilA* locus of *Streptococcus gallolyticus* can destabilize a premature transcription-terminating stem loop [61]. (**B**) Site-specific inversion is carried out by recombinases that recognize inverted repeat regions (Inverted Repeat Left/Right IRL/IRR) and flip the DNA sequence in between them. If a promoter region (e.g., p^B^) lies within the sequence flanked by the inverted repeats this leads to shut down of gene expression. (**C**) RecA-mediated DNA recombination of *N. gonorrhoea pilS* into *pilE* results in the formation of new *pilE* variants. Both *pilS* and *pilE* contain variable regions (depicted in green and orange, respectively) interspersed with conserved regions (white) while *pilE* has a further 5’ conserved region (dark orange) and a promoter to initiate transcription.

**Figure 2 microorganisms-09-00244-f002:**
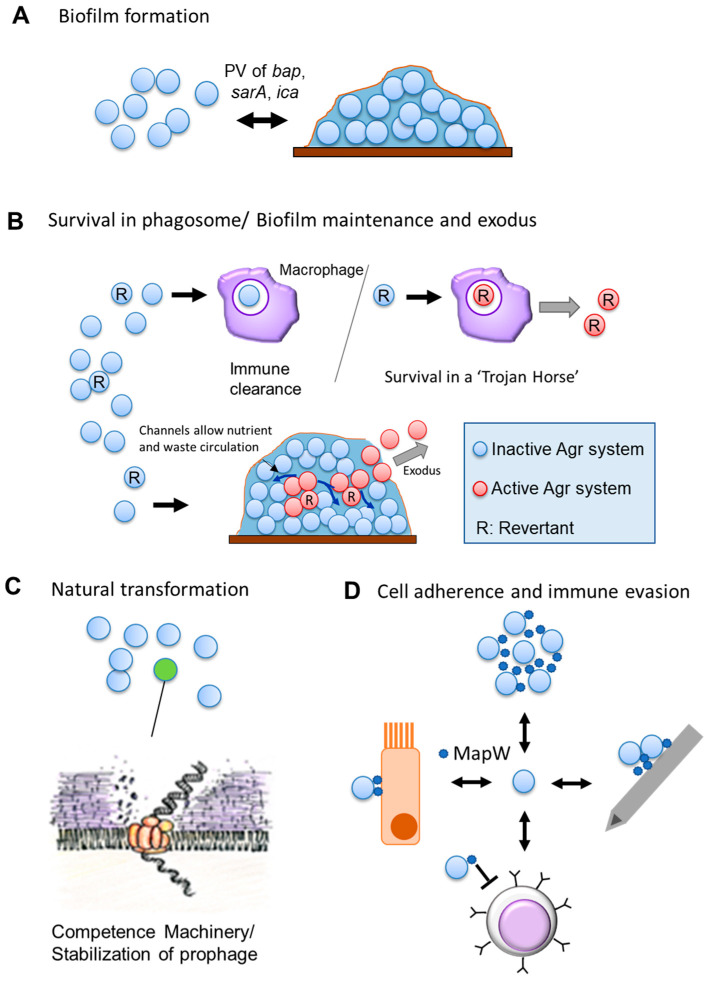
Phase Variation in *Staphylococcus aureus*. A cartoon depicting known PV in *S. aureus* and its roles. Double headed arrows indicate reversible PV events. (**A**) PV of Bap (Biofilm Associated Protein) (unclear mechanism) and of the *ica* (Intracellular Adhesion) operon (transposon insertion and SSR alteration) reversibly affects the biofilm-forming capability. (**B**) Phase variation of the Agr (Accessory Gene Regulator) system may have multiple possible roles [90]. It could serve as a cryptic insurance strategy against host immune attack, allowing phagocytosed revertant cells to activate their reverted Agr system to survive within a “Trojan Horse” (Top). It may also aid in the proper structuring of biofilms, as revertant cells and their progeny can activate their Agr system in structured non-planktonic architecture and the resultant exoproteins form channels in the biofilm for circulation of waste and nutrients as well as facilitating the exodus of cells from the biofilm [93] (Bottom). (**C**) PV-like expression is one of two independent expression mechanisms known for the alternative sigma factor σ^H^. Either mechanism allows for a subpopulation of cells to express the competence machinery and undergo natural transformation. (**D**) PV of the cell-wall associated MapW (MHC class II Analog Protein) may have multiple impacts [94]. MapW is involved in bacterial aggregation and may lead to biofilm formation; It is also implicated in the adherence to the host matrix and in-dwelling medical devices (e.g., cannulas); MapW has immunomodulatory effects and seems to suppress T-cells and their recruitment, though the exact mechanism remains elusive; MapW is important in adherence to, and internalization by, host non-phagocytic cells (e.g., epithelial cells). (Clockwise from top).

**Figure 3 microorganisms-09-00244-f003:**
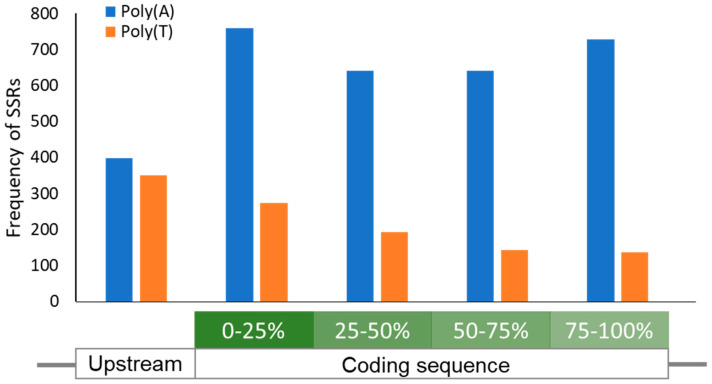
Location distribution of homonucleotide SSRs in *S. aureus*. A graph showing the location-dependent frequency of Poly(A)/(T) SSRs ≥6 nucleotides in length. A 100-nucleotide region upstream of all Open Reading Frames (ORFs) was also screened for SSRs. While Poly(T) SSRs are relatively enriched towards the 5’ of the ORF as reported [54], Poly(A) SSRs ≥6 nucleotides in length were found to be evenly distributed across the ORF. In contrast, the abundance of both SSR types is evenly matched in the region immediately upstream of the ORF.

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
