# Peer review of "No Change, No Life? What We Know about Phase Variation in Staphylococcus aureus"

_microorganisms, 2021, doi:10.3390/microorganisms9020244_

Round 1
Reviewer 1 Report
Dear Authors,
my best congratulations for your work, dealing with phase variation in St. aureus, which I found very interesting. I just have very few suggestions to improve your work:
1.
In the last part of introduction, you state that St. aureus is responsible for a range of infections. I think that it would be quite interesting to add few lines to list the main clinical pathologies caused by this bacterium. For example, it is related with bone-infections, implant-related infections, etc.
2.
Please add a reference after you state that PV can be discriminated into genetic and epigenetic mechanisms, as well as after you list the specific genetic mechanisms.
5.
- If you prefer, you could add at the beginning of this section (instead of at section 1) some example of clinical pathologies linked to St. aureus.
- “Mutator: relation with small colony variant, linezolid resistance, etc… Are these relate to SSM?” – I think that this sentence should be rephrased
6.
In the last lines of this section, I suggest rephrasing “. Here, we describe a comprehensive understanding” with “Here, we have described a comprehensive understanding”
Congratulations again for your work!
Reviewer 2 Report
Overall:
the paper is of interest to the journal as the concept of the PV is interesting, after major corrections
the concept of the paper is interesting, however, the authors need to work on bringing forth an overarching „message” of the review paper, in which they synthesize the currently available literature to put fort a „beginning” a core middle part and an „end” to the paper
Introduction:
the first Section of the MS is lacking in references. The authors should include relevant references and contextualize the text in accordance with the topic.
Hypermutability as an important aspect of genetic variability (e.g., in Pseudomonas in Cystic Fibrosis patients, or in S. aureus) should be briefly mentioned. Please consider including the following reference:
https://pubmed.ncbi.nlm.nih.gov/33406652/
Section 1. should be separated into a brief Introduction and a Section 2 where things of PV are discussed in more detail. The paragraph beginning with „The Gram-positive human commensal Staphylococcus aureus is an opportunistic path…” should be moved to the end of Section 1 and then continue on with PV in Section 2 (and in the other sections, respectively).
The introduction on the clinical relevance of S. aureus should be complemented and discussed more extensively.
Please consider including the following references:
https://pubmed.ncbi.nlm.nih.gov/31052511/
https://journals.sagepub.com/doi/full/10.1177/1179916117703999
Discuss that S. aureus is the member of the ESKAPE bacteria.
„The benefits of PV are not limited to pathogenic bacteria, however.” Please rephase this sentence
Nucleotide base names should not be capitalized.
With the beginnin of this sentence, a new section should be started:
„Just short of a decade ago, S. aureus was found to undergo natural transformation (72). These findings have implications regarding its rapid acquisition of antibiotic re-sistance genes, but intriguingly it was demonstrated that only a subset of the population can enter a competence state.„
„Mutator : relation with small colony variant, linezolid resistance, etc… Are these re-late to SSM?” either elaborate or delete
The Future perspectives section would benefit a lot from a summary table or figure. Please include it.
Round 2
Reviewer 2 Report
The authors have duly addressed my concerns regarding the manuscript. I commend the authors' committment to the improvement of the paper. In its current state, the manuscript is ready to be accepted.